# Investigation of Broadband Optical Nonlinear Absorption and Transient Dynamics in Orange IV Containing Azobenzene

**DOI:** 10.3390/molecules28124692

**Published:** 2023-06-10

**Authors:** Quanhua Wu, Rui Ruan, Xingxing Li, Yujie Zhao, Yang Li, Yu Fang, Yongqiang Chen, Quanying Wu, Yinglin Song, Xingzhi Wu

**Affiliations:** 1Jiangsu Key Laboratory of Micro and Nano Heat Fluid Flow Technology and Energy Application, School of Physical Science and Technology, Suzhou University of Science and Technology, Suzhou 215009, China; wuquanhua1998@163.com (Q.W.); 17774003013@163.com (R.R.); lixing17339649786@163.com (X.L.); 18361793147@163.com (Y.Z.); liyang@usts.edu.cn (Y.L.); yufang@usts.edu.cn (Y.F.); yqchen@usts.edu.cn (Y.C.); wqycyh@mail.usts.edu.cn (Q.W.); 2Department of Physics, Soochow University, Suzhou 215123, China; 3Department of Physics, Harbin Institute of Technology, Harbin 150001, China

**Keywords:** nonlinear optics, Z-scan, transient absorption and refraction, pump-probe

## Abstract

Broadband reverse saturable absorption is systematically investigated via Z-scan, transient absorption spectrum (TAS). The excited state absorption and negative refraction of Orange IV are observed in the Z-scan experiment at 532 nm. Meanwhile, two-photon-induced excited state absorption and pure two-photon absorption are observed at 600 nm and 700 nm with the pulse width of 190 fs, respectively. An ultrafast broadband absorption in the visible wavelength region is observed via TAS. The different nonlinear absorption mechanisms at multiple wavelengths are discussed and interpreted from the results of TAS. In addition, the ultrafast dynamics of negative refraction in the excited state of Orange IV are investigated via a degenerate phase object pump-probe, from which the weak long-lived excited state is extracted. All studies indicate that Orange IV has the potential to be further optimized into a superior broadband reverse saturable absorption material and also has certain reference significance for the study of optical nonlinearity in organic molecules containing azobenzene groups.

## 1. Introduction

The decades of progress in nonlinear optics research have been accompanied by the development of applications such as harmonic generation [1], broadband reverse saturation absorption [2,3], and all-optical switching [4]. For the realization of these unique applications, nonlinear optical materials are indispensable. Among them, broadband reverse saturable absorption materials are considered to have important application potential in the protection of the ultrashort pulse laser. Thus, organic nonlinear materials are attracting widespread concern because of their large optical nonlinearity, broad spectral response, and easy modification. The optical nonlinear response of organic materials is decisively related to the off-domain π-electrons within the π-conjugated system of the molecule. The azobenzene group has a fast nonlinear optical response and a large nonlinear optical (NLO) coefficient due to its large π-conjugated system [5,6,7,8]. The NLO properties of the azobenzene group have broad application prospects in photorefractive, optical communication, and magnetic memories materials [9,10,11]. Meanwhile, azobenzene and its derivatives, due to their trans and cis isomerization, thus constitute a series of important photonic switches [12,13,14,15]. The cis-trans structural isomerization, furthermore, may have induced changes in the third-order NLO behavior of the molecule [16,17]. A previous report showed an enhanced NLO response at 1064 nm for some azobenzene derivative molecules associated with the cis-isomer [18]. Other reports have investigated the ability of polymer films and complexes of some azobenzene derivatives to induce two-photon absorption based on a nonlinear optical-photo isomerization cycle model [19,20]. These results demonstrate the value of azobenzene-containing organic molecules for applications in nonlinear optics research. However, most of the studies focus on the photoisomerization process of azobenzene and its derivatives, with little attention paid to the broadband nonlinear absorption and transient refraction of molecules containing azobenzene groups. The azobenzene group has a fast nonlinear optical response and a large nonlinear optical (NLO) coefficient due to its large π-conjugated system. By introducing different electron-withdrawing and electron-donating groups into the aromatic ring at both ends, the internal charge mobility of the molecule can be increased under the action of an electric field.

Here, in this work, the broadband nonlinear optical absorption of an azobenzene-containing molecule (Orange IV) at multiple wavelengths is investigated by femtosecond Z-scan, and results indicate that the sample exhibits reverse saturation absorption (RSA) at 532 nm, and two-photon-induced excited state absorption (TP-ESA) is observed at 600 nm, while two-photon absorption (TPA) is observed at 700 nm. The different nonlinear absorption mechanisms at various wavelengths are successfully verified and explained by TAS, which is in good agreement with the results of femtosecond Z-scans at multiple wavelengths. In addition, RSA and negative refraction of the excited state are observed at 532 nm with various pulse width lasers (including 190 fs, 13 ps, 4 ns). The ultrafast dynamics of negative refraction in the excited state of Orange IV are investigated via a degenerate phase object pump-probe, from which the weak long-lived excited state is extracted. These results provide some reference to the research of the optical nonlinearity of some organic molecules containing azobenzene.

## 2. Results and Discussion

### 2.1. UV-Vis Absorption and Fluorescence

We monitored the UV-vis absorption and fluorescence spectrum of Orange IV at room temperature. The excitation wavelength of the fluorescence spectrum is 380 nm. As shown in Figure 1, the strong visible absorption band arises at the wavelength of 436 nm. The fluorescence emission peak corresponding to the absorption peak appears at the 591 nm wavelength in the detection window. Comparing the UV-vis and fluorescence spectrum, we can find that the absorption and emission peaks agree with the mirror image rule, indicating that the absorption signal may originate from the first singlet state [21,22,23].

In order to obtain more linear absorption information, DFT calculations are conducted as shown in Figure 2. It can be found that there is a visible charge transfer in the right benzene ring under excitation. The calculated energy gap between HOMO and LUMO is 2.921 eV, which corresponds to a wavelength of 425 nm, very close to the absorption peak of Orange IV solution at 436 nm. These results indicate that the sample has a visible intramolecular charge transfer (ICT) feature.

### 2.2. The Third-Order NLO Research

To evaluate the third-order optical nonlinearity of Orange IV, Z-scan measurements are conducted under various pulse widths. The calibration of the extracted data can be significantly improved by using 3 mm thickness zinc selenide as a standard sample. The Z-scan with open aperture (OA) reflects the nonlinear absorption of the sample, while the closed aperture (CA) Z-scan reflects the nonlinear refraction of the sample. Femtosecond Z-scan experiments were conducted at 532 nm, 600 nm, and 700 nm, respectively. When excited with 532 nm laser pulses (incident laser intensity of 21.4 GW/cm^2^), the OA Z-scan (Figure 3a) of Orange IV displays a single valley at zero position (focal point), indicating the lower transmittance under higher laser intensity, which is a typical feature of reverse saturable absorption. Nonlinear absorption and nonlinear refraction of the solvent and solution are measured with the same laser irradiation; their nonlinear absorption coefficient and refractive index are then extracted from numerical simulation. The nonlinear absorption coefficient of the solution combined with the nonlinear refractive index of the solvent are used for another simulation, which represents the pure refractive effect of the solvent in our solution. This refractive background is then removed from the solution to obtain the nonlinear refractive signal of the solute. The CA Z-scan curve (Figure 3b) of the sample after the removal of the solvent background (as shown in Appendix A) displays the shape of a peak followed by a valley, suggesting a negative change of the nonlinear refractive index as the sample moves from the −z to +z position, referred to as the self-defocusing effect. Considering that 532 nm is located within the edge of the linear absorption band and the linear transmittance of the sample is 0.35 (excluding the reflection of the cuvette), both the RSA and the negative nonlinear refraction (NLR) under resonant excitation could be attributed to the excited state optical nonlinear response.

When tuning the laser incident wavelength to a non-resonant absorption band, Orange IV exhibits an extremely high linear transmittance of 0.93 at 600 nm and 700 nm (almost identical to linear transmittance of the pure solvent DMSO). The OA Z-scan curves of Orange IV at both 600 nm and 700 nm similarly record a single valley, as shown in Figure 3c,d. Unlike the case of 532 nm, as the 600 nm and 700 nm wavelengths are away from the resonant absorption band, the nonlinear absorption (NLA) at both wavelengths cannot originate from single photon-induced excited state absorption; therefore, we assign the NLA to multi-photon absorption. It should also be noted that after eliminating the effect of positive refraction from the solvent background, NLR was not observed in Orange IV solution at both wavelengths, which is different from the negative refraction at 532 nm.

Numerical simulations based on Sheik Bahae’s theory are used to fit the results from the Z-scan experiment, where the nonlinear absorption coefficient and refractive index of the sample are expressed as:(1)$$\alpha =\alpha_{0}+\beta I+(\gamma I^{2})$$
(2)$$n=n_{0}+n_{2}I$$
where *α*_0_ is linear absorption coefficient, *β* and *γ* are effective nonlinear absorption coefficients for third-order and fifth-order, respectively. If only third-order nonlinear absorption exists in the sample, *γ* is zero. Similarly, *n*_0_ and *n*_2_ represent the linear refractive index and third-order nonlinear refractive index, respectively. The variation of laser intensity (I) and phase (φ) caused by nonlinear light absorption and refraction in the sample can be described as:(3)$$\frac{dI}{dz}=-\alpha I$$
(4)$$\frac{d\varphi }{dz}=\frac{2\pi }{\lambda }nI$$
where *z* stands for propagation depth in the sample. According to the above theory, the values of these parameters were obtained by numerical simulation, solid lines in Figure 3.

As discussed above, the optical NLA and NLR at 532 nm wavelength are mainly determined by excited state absorption and refraction. As for the 600 nm case, at different peak intensities including 21.7 GW/cm^2^, 27.6 GW/cm^2^, and 34.4 GW/cm^2^, the corresponding third-order nonlinear coefficients *β* are 1.3 × 10^−13^ m/W, 1.6 × 10^−13^ m/W, and 2.0 × 10^−13^ m/W, respectively, which are almost proportional to the incident intensity. This suggests that there may be a higher order NLA at 600 nm. The *γ*(*I*^2^) term is then additionally taken into account in our absorption coefficients, resulting in third- and fifth-order nonlinear absorption coefficients *β* and *γ* independent of the incident intensity (listed in Table 1), which matched the experimental data well. It is noted that the 600 nm wavelength is in the non-resonant absorption band, but it is unlikely to absorb three photons at this wavelength at the same time. Therefore, the RSA is more likely to originate from TP-ESA. However, the NLA was observed at an incident intensity of 41.8 GW/cm^2^ when the sample was excited at 700 nm, and effective NLA signals cannot be extracted from the noisy background at lower intensities. Noting that the energy of two 700 nm photons is equivalent to the energy of a single photon at 350 nm, and yet the linear absorption at 350 nm is weaker as seen in Figure 1, thus the NLA here can be considered to be purely dominated by TPA. The nonlinear parameters for the different wavelengths in the femtosecond Z-scan are listed in Table 1.

Furthermore, as shown in Figure 4, picosecond and nanosecond Z-scans are also implemented at 532 nm with incident laser intensities of 2.7 GW/cm^2^ and 38.2 MW/cm^2^. The OA Z-scan of Orange IV with ps and ns pulses shows a single valley at focal point (the zero position), indicating lower transmittance at higher laser intensities, a typical feature of reversed saturable absorption. The CA Z-scan curves also display the shape of peak-valley under the same energy, indicating negative nonlinear refraction, namely, the self-defocusing effect. We then fitted the corresponding nonlinear absorption and refraction parameters, as shown in Table 2. Similar to the case in the 532 nm femtosecond Z-scan, the Orange IV molecule also exhibits single photon-induced ESA and negative refraction.

### 2.3. Transient Absorption Spectrum

The results of the Z-scan indicate that different absorption mechanisms play a key role in the optical nonlinearity of Orange IV. To further distinguish and explain the differences of the NLA at various incident laser wavelengths, transient absorption spectrum is conducted in the visible region.

The TAS is recorded and displayed in Figure 5a with the probe window ranging from 505–750 nm. In the TAS, ΔOD represents the variation of optical density with the definition ΔOD *=* −log (*I_pnmped_*/*I_unpumped_*), where *I_pnmped_* and *I_unpumped_*, respectively, represent the intensity of the probe pulse on the detector with and without pumping. In addition, ΔmOD *=* 10*^−^*^3^ ΔOD. It is clear that the broadband RSA of the sample covers the entire detection wavelength region. In addition, for more details, we extract spectral curves at different delay times, which are shown in Figure 5b. The recorded ultrafast broadband excited state absorption builds up within a pulse width. Moreover, the excited state absorption (ESA) signal increases in the range of approximately 505–630 nm and decreases in the range of 630–750 nm as the absorption spectrum with a time delay of 0.2 ps evolves to that of 0.4 ps. After that, the spectral curve decays in a similar shape within a very short time. Considering the spectrum curve with a lifetime of 0.2 ps (about one pulse width) and femtosecond Z-scan, the excited state absorption is found at 532 nm and 600 nm, which explains the existence of single photon-induced excited state absorption and TP-ESA at 532 nm and 600 nm, respectively, while the ESA signal at 532 nm is stronger than that at 600 nm, which is consistent with the results of the femtosecond Z-scan. Furthermore, the excited state absorption found at 700 nm is comparatively little, which may be the reason that TPA at 700 nm could not trigger the subsequent excited state absorption as it could at 600 nm.

The global analysis [24] is used to numerically reconstruct the TAS. The TAS could be divided into different components for a reasonable fit, and each component has a featured absorption spectrum (*A_i_*) and a corresponding decay lifetime. Thus, the entire TAS could be expressed as:(5)$$\begin{array}{l} \Delta T(t)=\sum_{i=1,n}\int_{-\infty }^{+\infty }R_{IRF}(\tau )[A_{i}\exp(-\frac{t-\tau }{\tau_{i}})]d\tau \\ =R_{IRF}⊗[\sum_{i=1,n}A_{i}\exp(-\frac{t}{\tau_{i}})] \end{array}$$
where Δ*T*(*t*) stands for the normalized transient absorption signal and *R_IRF_* represents instrument response curve. Parameters *τ_i_* and *A_i_* are time and amplitude terms of the exponential decay function, respectively. ⊗ denotes the convolution operation. We used multi-e-exponential functions to fit the photodynamic curves collected at different probe wavelengths. The lifetimes of a total of two components are used to reasonably fit the TAS. The spectral components extracted from the global analysis are shown in Figure 5c. The lifetimes of the two components extracted from the simulations are 0.3 ps, 4.1 ps, respectively. The numerical simulation results of the dynamics for the corresponding wavelengths used in the Z-scan are plotted in Figure 5d. The *x*-axis of Figure 5d is not completely linear. The ultrafast process with a lifetime of 0.3 ps lifetime is assigned to ultrafast cooling of the locally excited state (LE). It then relaxes to other states in the molecule with a lifetime of 4.1 ps. The process with a lifetime of 4.1 ps may involve a relaxation from the LE state to the charge transfer state (CT).

### 2.4. Transient Refraction

It is also interesting to note that Orange IV still exhibits negative NLR from the excited state in the Z-scan results under 532 nm. Since negative nonlinear optical refraction could be applied to all-optical switching [25,26], the dynamics of NLA and NLR are investigated via a degenerate phase object pump-probe (POPP) technique. POPP technology enables simultaneous measurement of NLA and refraction by transforming nonlinear refractive index changes into transmittance changes via phase objects and can detect transmittance changes with an accuracy of 0.02. The sample solution is pumped and probed at 515 nm. The POPP results are shown in Figure 6.

The NLA dynamics in Figure 6a show a strong RSA at zero time followed by a decrease with the lifetime of 0.3 ps and 4.1 ps, which is in good agreement with the kinetic results in the TAS. Moreover, the long-lived weakly excited state absorption (normalized transmittance of about 0.98) is extracted in Figure 6c, which indicates that the absorption cross section of the excited state is slightly larger than that of the ground state (S_0_). This explains the OA nanosecond Z-scan results. Correspondingly, the dynamics of the NLR (Figure 6b,d) begin with an ultrafast rise in positive refraction near zero time, which originates from the Kerr refraction of the solvent background. After that, the dynamics of NLR display an ultrafast drop, which indicates the strong negative excited state NLR in Orange IV solution. This is immediately followed by a fast recovery, including two decay processes with lifetimes of 0.3 ps and 4.1 ps, which agree with the results in the transient absorption spectrum and correspond to the relaxation of the local excited state and charge transfer state, respectively. Finally, there is a weak negative signal in the transient refraction and it maintains for a relatively long time, corresponding to a long-lived weakly excited state energy level (CT). These results are also consistent with the above discussion of Z-scan with multiple pulse widths.

Generally, after excitation by the pump pulses, the excited state electrons decay to each excited state in turn and finally relax to the ground state. Simplified rate equations based on the energy model are used to numerically fit the results. The modulation of the intensity (*I_p_*) and phase (*φ_p_*) of the probe beam can be expressed as:(6)$$\frac{dI_{p}}{dz}=-I_{p}(\sum_{n\geq 1}\sigma_{n}N_{n}+2\beta I_{e})$$
(7)$$\frac{d\varphi_{p}}{dz}=k(\sum_{n\geq 1}\Delta \eta_{n}N_{n}+2n_{2}I_{e})$$
where *I_p_* and *I_e_* represent the light intensity of the probe beam and the pump beam, and *n*_2_ is the Kerr refractive index related to the incident intensity, which is set to *n*_2_ = 6.5 × 10^−19^ m^2^/W based on the experiment. Furthermore, *z* represents the propagation length of the laser in the sample, *σ_n_* stands for the absorption cross section of the excited state, and Δ*η_n_* is the change of refractive volume between the effective energy state and the ground state. As discussed above, the equivalent energy level model as shown in Figure 7 is built up to fit the data. Following numerical simulations, a series of parameters including excited state absorption cross sections, refractive volumes, and lifetimes are fitted and listed in Table 3. It is worth noting that the lifetimes of CT state larger than 4 ns cannot be determined accurately due to the limited delay time.

## 3. Materials and Methods

### 3.1. Characterization of Orange IV

As a general biological stain and acid-base indicator, the Orange IV is purchased from commercial vendors and used directly. The structure of Orange IV is shown in Figure 1. Additionally, 1H NMR (400 MHz, DMSO-d6) δ 8.89 (s, 1H), 7.82 (d, J = 8.9 Hz, 2H), 7.74 (s, 4H), 7.37–7.29 (m, 2H), 7.22 (d, J = 7.3 Hz, 2H), 7.17 (d, J = 9.0 Hz, 2H), 6.99 (t, J = 7.3 Hz, 1H).

### 3.2. Z-Scan Experiment

The open and closed aperture Z-scans are conducted to measure the nonlinear absorption and refractive response of Orange IV [27]. The laser source included a mode-locked Yb: KGW-based fiber laser (1030 nm, Orpheus, Light Conversion) with an output pulse width of 190 fs, a Q-switched and mode-locked Nd: YAG laser (PW-1064-1B) working at 532 nm with 13 ps pulse width (FWHM), and 4 ns (FWHM) pulse of 532 nm were extracted from a Q-switched Nd: YAG laser (Surelite II, Continuum). The low repetition rate is set to 20 Hz in femtosecond and picosecond pulses and 10 Hz in nanosecond pulse to avoid thermal lens effect caused by heat accumulation of the sample [28,29]. The Orange IV is dissolved in DMSO with the concentration of 2.66 × 10^−3^ M and contained in a 2 mm quartz cuvette. The sample solution is placed on an electric translation platform, which is controlled by the set program and moves along the z-axis. During the movement of the sample, energy detectors are used to collect the energy changes in the transmitted beam as it moves along the z-axis. An aperture is placed in front of one of the energy detectors. When the aperture is completely open, the energy of the transmitted beam is recorded as an Open-Aperture curve, and the information of the nonlinear absorption in the sample is obtained. When the aperture is closed, in this case, the transmittance of the closed aperture is about 0.25; the data curve recorded by the detector is then divided by the open aperture z-scan curve to obtain the closed aperture curve, which provides information on the nonlinear refraction.

### 3.3. Femtosecond Transient Absorption Spectroscopy

The transient absorption spectrum can record the ultrafast absorption and kinetic relaxation processes at different wavelengths after excitation. The pump beam of TAS is extracted from an optical parametric amplifier (OPA) (ORPHEUS, Light Conversion), which is pumped via a 1030 nm femtoseconds laser system (Pharos, Light Conversion). The wavelength of the pump beam could be tuned in a wide spectral range. Femtosecond transient absorption spectroscopy is an optimized investigation tool based on the traditional pump-probe technique. The output of the laser is divided into two beams, one as pump beam and one as probe beam. The probe beam is focused into a sapphire crystal and transformed into a super-continuous white light. Compared with the traditional pump detection technique with a single detection wavelength, the transient absorption spectra can provide more details of dynamics of excited states [30]. The femtosecond fiber laser used in the experiment is the same as in the Z-scan above, and the pump pulse is tuned to 400 nm (14 mW). The repetition rate of the laser pulse is 6 kHz. The probe window of the grating spectrometer is adjusted to 505–750 nm. The sample solution is contained in a 2 mm cuvette.

### 3.4. Phase Object Pump-Probe

The dynamic traces of ultrafast nonlinear absorption and refraction are conducted by the pump-probe technology with phase object [31]. The phase object pump-probe (POPP) technology is a combination of a conventional pump-probe and a 4f phase object imaging system [32]. The difference between the POPP technique and traditional pump-probe is that POPP can simultaneously investigate the dynamic of nonlinear absorption and refraction conveniently. By introducing a phased object, the phase shift caused by nonlinear refraction can be modulated to a change in intensity inside the phase object area. If the nonlinear refractive index of the sample is positive, the laser spot will have an increased intensity inside the phase object area. Conversely, if the nonlinear refractive index of the sample is negative, the laser spot will have a weakened intensity inside the phase object area. The laser source used in the experiment is the same as in the TAS. The repetition frequency of the laser pulse is 20 Hz. The wavelengths of the pump and probe light of degenerate POPP are both 515 nm. The sample dissolved in DMSO with a concentration of 6.65 × 10^−4^ M is prepared in a 2 mm quartz cuvette for measurement.

### 3.5. Quantum Chemical Calculations

Quantum chemical calculations are performed on Orange IV at the level of B3LYP/6-31G (d,p) via the Gaussian 09 program package to obtain optimized structures and frontier molecular orbitals.

## 4. Conclusions

The broadband optical nonlinear absorption of Orange IV is investigated systematically. The strong ESA and nonlinear negative refraction at 532 nm, the TP-ESA at 600 nm, and the TPA at 700 nm are observed via the femtosecond Z-scan. Moreover, picosecond and nanosecond Z-scans also reveal that the molecules have excited state absorption and negative refraction. Ultrafast broadband excited state absorption is observed in the TAS, and the NLA mechanisms of different wavelengths are verified and characterized. In addition, the long-lived weak excited state of nanosecond scale is observed, and the negative refraction dynamics of excited state response are extracted in POPP experiments, which is consistent with the results of Z-scan with multiple pulse widths. These studies suggest that Orange IV has the potential to further optimize into superior broadband reverse-saturated absorption materials, and it has certain reference significance for the study of optical nonlinearity in organic molecules containing azobenzene.

## Figures and Tables

**Figure 1 molecules-28-04692-f001:**
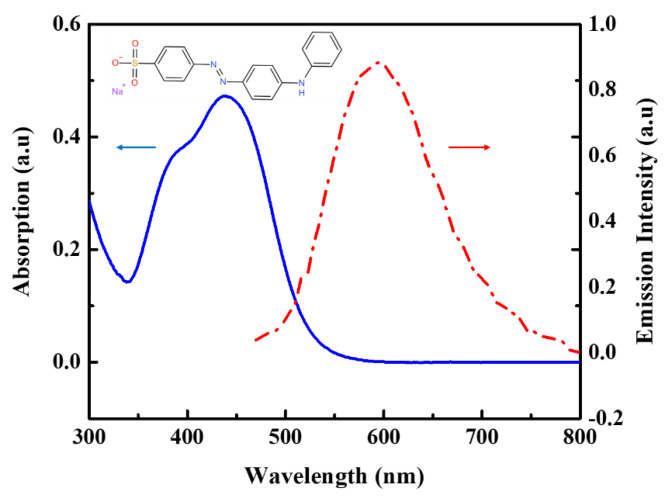
UV-vis absorption (blue) and fluorescence (red) spectra of Orange IV. The concentrations of Orange IV are 2.66 × 10^−5^ M in dimethyl sulfoxide (DMSO) solution.

**Figure 2 molecules-28-04692-f002:**
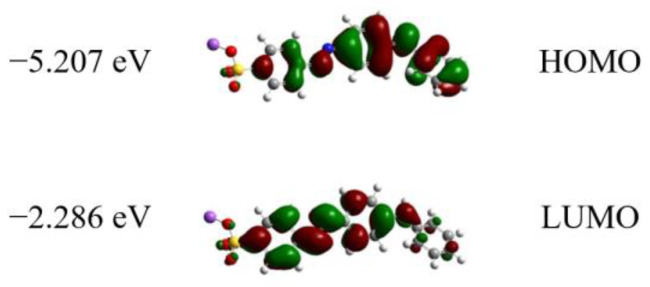
Electron distribution and energy of frontier molecular orbitals in Orange IV.

**Figure 3 molecules-28-04692-f003:**
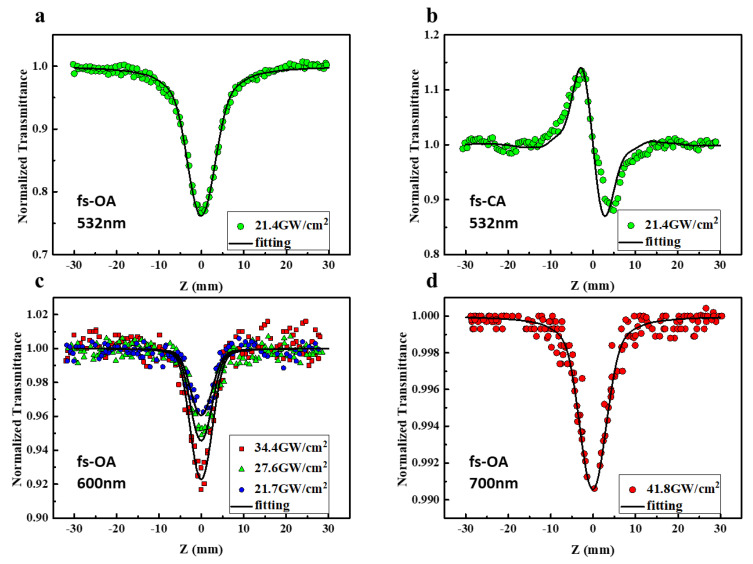
Femtosecond Z-scan measurement curve of Orange IV. (**a**) Open aperture curve at 532 nm. (**b**) Closed aperture curve at 532 nm. (**c**,**d**) Open aperture curve at 600 nm and 700 nm. Solid line represents the results of numerical fitting.

**Figure 4 molecules-28-04692-f004:**
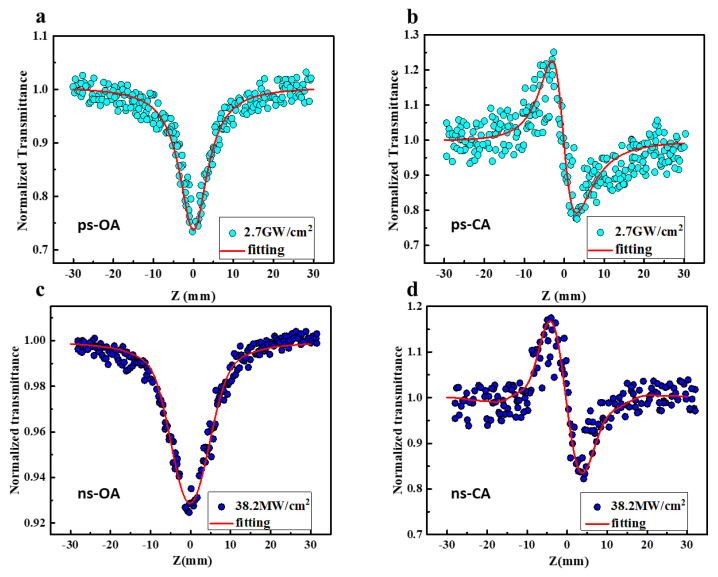
The open and closed aperture picosecond (**a**,**b**) and nanosecond (**c**,**d**) Z-scan curves of Orange IV under 532 nm. The solid line represents the theoretical fitting.

**Figure 5 molecules-28-04692-f005:**
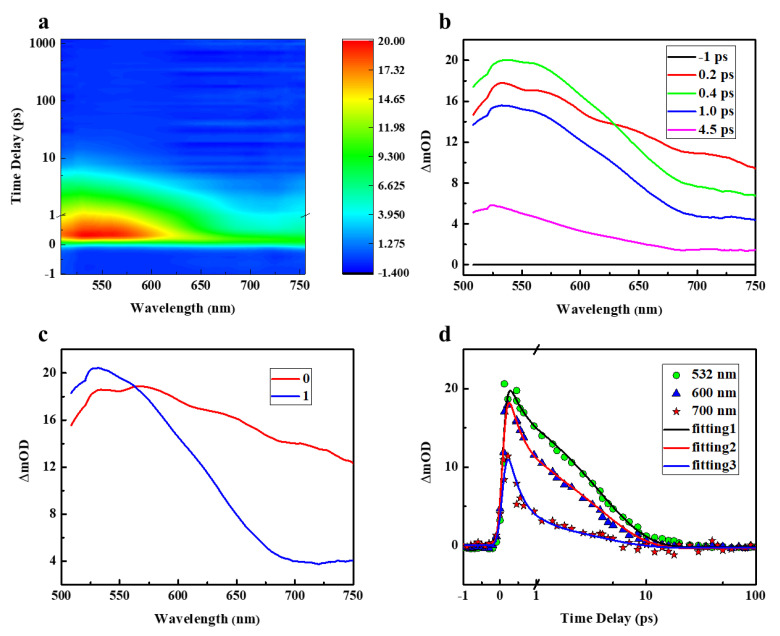
Results of transient absorption spectra of Orange IV solutions. (**a**) A 2D color map of transient absorption experiment. (**b**) TAS under different delay times. (**c**) Spectral components extracted from global analysis. (**d**) Numerically fitted kinetic curves at various wavelengths.

**Figure 6 molecules-28-04692-f006:**
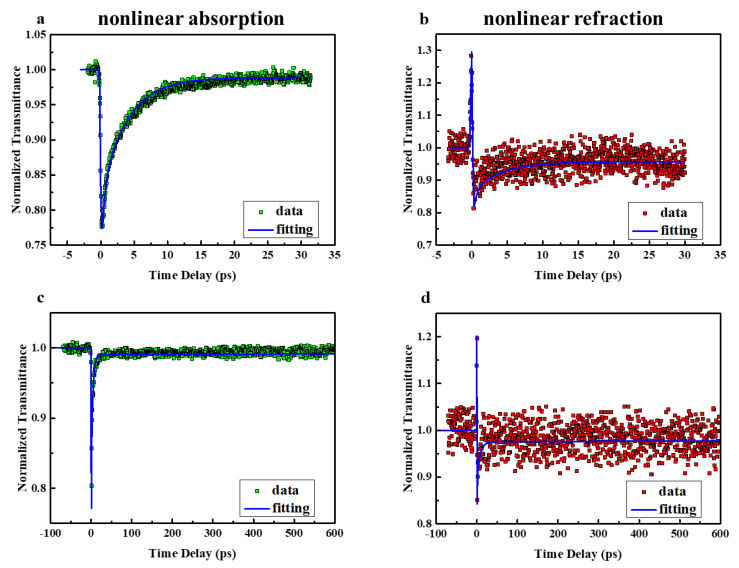
The transient (**a**,**c**) NLA and (**b**,**d**) NLR of Orange IV probed at 515 nm, measured with degenerate POPP. Solid lines represent the results of the numerical simulation.

**Figure 7 molecules-28-04692-f007:**
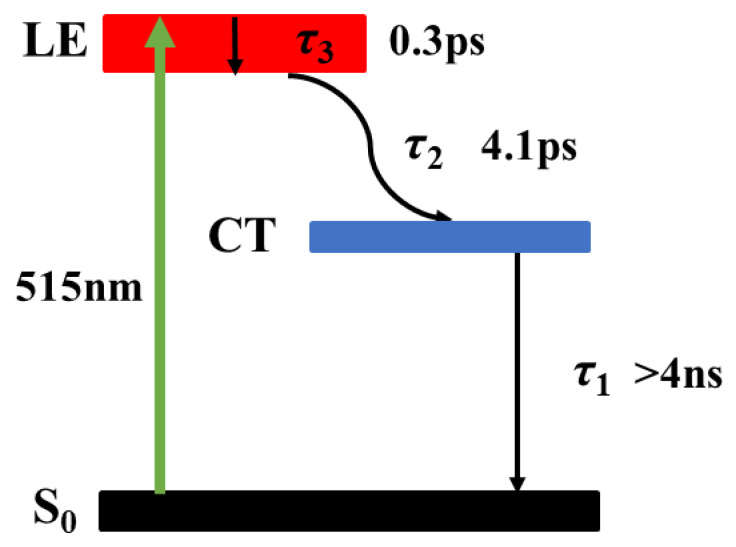
Simplified energy level of the excited state of Orange IV.

**Table 1 molecules-28-04692-t001:** Nonlinear parameters of femtosecond Z-scan experiment.

	532 nm, T = 0.35	600 nm, T = 0.93	700 nm, T = 0.93
Orange IV	*β* = 2.2 × 10^−12^ m/W	*β =* 4.4 × 10^−14^ m/W	*β* = 2.5 × 10^−14^ m/W
	*n*_2_ = -6.5 × 10^−20^ m^2^/W	*γ* = 6.3 × 10^−28^ m^3^/W^2^	

**Table 2 molecules-28-04692-t002:** Nonlinear parameters of picosecond and nanosecond Z-scan experiment.

	13 ps, T = 0.35	4 ns, T = 0.35
Orange IV	*β* = 1.6 × 10^−11^ m/W	*β =* 3.8 × 10^−10^ m/W
	*n*_2_ = 2.5 × 10^−18^ m^3^/W^2^	*γ* = 6.3 × 10^−17^ m^3^/W^2^

**Table 3 molecules-28-04692-t003:** Parameters of excited states extracted from the numerical simulation.

State	*σ_n_* (m^2^)	Δ*η_n_* (m^3^)	*τ_n_* (ps)
LE	8.8 × 10^−21^	-3.1 × 10^−21^	0.3
	4.7 × 10^−21^	-1.1 × 10^−21^	4.1
CT	1.4 × 10^−21^	-7.2 × 10^−22^	>4000
S_0_	1.2 × 10^−21^	0	-

## Data Availability

Not applicable.

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
