# Peer review of "Investigation of Broadband Optical Nonlinear Absorption and Transient Dynamics in Orange IV Containing Azobenzene"

_molecules, 2023, doi:10.3390/molecules28124692_

Round 1
Reviewer 1 Report
The authors present a study of the nonlinear absorption and refraction on a solution of Orange IV, an azobenzene. the results seem interesting enough for publication, however, there are some points that need to be addressed before the manuscript can be accepted. These are the points:
- Is the molecule Orange IV commercially available? Or was it synthesized by the authors? This must be clearly stated
- In section 2.2 regarding the z-scan results, the authors mention that the nonlinear refractive index of the solvent is removed from the results to obtain that of the solute. How is this achieved? A detailed explanation is needed. The supplementary material shows something, but is not enough.
- For the results at different wavelengths the authors fit the open-aperture z-scan results with a model considering in some cases only TPA and in others three-photon absorption. They do not show how do the data require in one case only 2PA, and in others 2 and 3PA, this must have to do with the shape of the open-aperture trace. This can also be observed from a graph of fitted beta vs input irradiance for data at different input irradiances. In one case (only beta) there won't be a dependence, while in the other, a linear dependence of beta should be expected.
- the authors should show the expressions used to fit the z-scan results, including the ones where higher order terms are included.
- The authors never discuss the results shown in fig. 4 for ps and ns pulses at 532 nm.
- Regarding the transient absorption studies, the experimental data is not properly presented in fig. 5. Fig 5 seems to present all the results with the absorbance change coded in color, the x axis being the wavelength, and the y axis the probe delay time, not Delta mOD as it shows. This a major problem that makes understanding the results very difficult. The authors also should define somewhere what do they mean by Delta mOD, I assume that is the change in optical density (OD) in 1/1000 units, but it needs to be mentioned explicitly. Fig. 5d shows the usual results for individual wavelengths together with some fits, which expressions are used to obtain these fits? They don't have a purely exponential shape. The authors claim that they come from expression (3) , but too little detail is given.
-The authors then present results for the dynamics of the refractive nonlinearity, employing a phase-object pump-probe technique that is not adequately described in the manuscript. They then use what it seems a multilevel rate equation system to explain the results obtained, something that is not clear at all.
- The authors assign the observed response times to excited states lifetimes, which are in the order of ps, and they assign an observed 4 ns lifetime to the vibrational decay within the electronic ground state! This vibrational decay is usually very fast, of the order of 1 ps or faster. There is clearly something wrong in this assignment.
- This assignment of lifetimes is confusing, in the paragraph above fig. 5 the authors state '...The slower component with a lifetime of 4.1 ps is attributed to the relaxation of the CT state, and the slower decay lifetime may be related to the photoisomerization process in terms of previous reports [26-27]..' They seem to be assigning one lifetime to two different processes. In fact photo-isomerization, that has been shown to have an important role in the nonlinear properties of azobenzenes, does not seem to be relevant in their whole explanation.
For this reason, the manuscript cannot be accepted in its present form
English grammar can be improved slightly, but in general is quite good.
Reviewer 2 Report
The paper submitted by Y.Song and X.Wu deals with systematically investigation of nonlinear properties of Orange IV. The methodology setup of study is completely correct and well made. However, the hurtful mistakes are occurred and it can’be be published as is. I recommend to accept the manuscript after minor revision, the issues are below. I believe that article is interest of readers of Molecules.
Major issues:
1. The azo-benzene dyes are well known and wide spreader. The choice of studied object should be justified and indicated in introduction section, because is not clear why Orange IV was chosen instead of other azo-benzene dyes.
Minor issues:
1. Figure 1, the excitation wavelength for recorder emission spectra should be indicated
2. Line 79 «2.912ev» the space is missed, should be «2.912 ev»
3. To make the conception more clearify for readers, I recommend to flip horizontal the structure on figure 1. Due to on figure 2 the N-phenyl moiety on the left and the authors claimed “left benzene ring”, but on figure 1 one is on the right side.
4. Line 90 spaces is missed between value and unit, i.e. “532nm, 600 nm…” etc.
5. Line 94 and Line 95 and Line 124 and Line 140 and Line 144 and Line 201the same issues on point 4 for “Figure3b” and “FigureS1” and “Figure3” and “Figure1” and “Figure 4” “Figure6”
6. Line 244 “Figure1” missed space and 1H NMR “1” should be as superscript
Author Response
Dear Reviewer,
Thank you very much for your supervision of the reviewing process of our manuscript (, Investigation of broadband optical nonlinear absorption and transient dynamics in Orange IV containing azobenzene). We also highly appreciate the referee’s conscientious, carefulness, and the broad knowledge on the relevant research fields since they have given us several beneficial suggestions. According to the referee's reports, we have made the following revisions on this manuscript:
Reviewer 2
Comments and Suggestions for Authors
The paper submitted by Y.Song and X.Wu deals with systematically investigation of nonlinear properties of Orange IV. The methodology setup of study is completely correct and well made. However, the hurtful mistakes are occurred and it can’be be published as is. I recommend to accept the manuscript after minor revision, the issues are below. I believe that article is interest of readers of Molecules.
Major issues:
- The azo-benzene dyes are well known and wide spreader. The choice of studied object should be justified and indicated in introduction section, because is not clear why Orange IV was chosen instead of other azo-benzene dyes.
Reply:Thanks for your valuable comment. The azobenzene group has a fast nonlinear optical response and a large nonlinear optical (NLO) coefficient due to its large π-conjugated system. By introducing different electron-withdrawing and electron-donating groups into the aromatic ring at both ends, the internal charge mobility of the molecule can be increased under the action of electric field.
Minor issues:
- Figure 1, the excitation wavelength for recorder emission spectra should be indicated
Reply:Thanks for the comment. The excitation wavelength is 380 nm.
- Line 79 «2.912ev» the space is missed, should be «2.912 ev»
- To make the conception more clearify for readers, I recommend to flip horizontal the structure on figure 1. Due to on figure 2 the N-phenyl moiety on the left and the authors claimed “left benzene ring”, but on figure 1 one is on the right side.
- Line 90 spaces is missed between value and unit, i.e. “532nm, 600 nm…” etc.
- Line 94 and Line 95 and Line 124 and Line 140 and Line 144 and Line 201the same issues on point 4 for “Figure3b” and “FigureS1” and “Figure3” and “Figure1” and “Figure 4” “Figure6”
- Line 244 “Figure1” missed space and 1H NMR “1” should be as superscript
Reply:Thanks for the comment. We have carefully checked and corrected some grammatical and spelling mistakes.
Round 2
Reviewer 1 Report
The authors have answered most of the points raised on the first review. Some things can still be improved :
- Some sort of rate equations is presented (eqs. 6-7) but it is not clear how are they used in the context of the experimental results.
- It is not clear what role us played, if any, by the photoisomerization process that can be present in the material.
The English language is OK
